# Predicting Apoptosis Protein Subcellular Locations based on the Protein Overlapping Property Matrix and Tri-Gram Encoding

**DOI:** 10.3390/ijms20092344

**Published:** 2019-05-11

**Authors:** Yang Yang, Huiwen Zheng, Chunhua Wang, Wanyue Xiao, Taigang Liu

**Affiliations:** 1AIEN Institute, Shanghai Ocean University, Shanghai 201306, China; 1591117@st.shou.edu.cn (Y.Y.); poppy_xiaowanyue@163.com (W.X.); 2College of Sciences & Engineering, University of Tasmania, 7001 Tasmania, Australia; huiwenz@utas.edu.au; 3College of Information Technology, Shanghai Ocean University, Shanghai 201306, China; wchshou@163.com; 4Key Laboratory of Fisheries Information Ministry of Agriculture, Shanghai 201306, China

**Keywords:** tri-gram, protein overlapping property matrix, subcellular location, support vector machine, recursive feature elimination

## Abstract

To reveal the working pattern of programmed cell death, knowledge of the subcellular location of apoptosis proteins is essential. Besides the costly and time-consuming method of experimental determination, research into computational locating schemes, focusing mainly on the innovation of representation techniques on protein sequences and the selection of classification algorithms, has become popular in recent decades. In this study, a novel tri-gram encoding model is proposed, which is based on using the protein overlapping property matrix (POPM) for predicting apoptosis protein subcellular location. Next, a 1000-dimensional feature vector is built to represent a protein. Finally, with the help of support vector machine-recursive feature elimination (SVM-RFE), we select the optimal features and put them into a support vector machine (SVM) classifier for predictions. The results of jackknife tests on two benchmark datasets demonstrate that our proposed method can achieve satisfactory prediction performance level with less computing capacity required and could work as a promising tool to predict the subcellular locations of apoptosis proteins.

## 1. Introduction

Apoptosis, as a form of programmed cell death occurring in multicellular organisms, is vital for balancing cell proliferation and death. Malfunction of apoptosis can trigger undesirable maladies including cancer, autoimmune disease, ischemic damage, and neurodegenerative disease [1]. It has been certified that the apoptosis proteins’ functions are closely connected with their subcellular locations [2]. Hence, assigning subcellular locations for apoptosis proteins is a crucial step to understanding their working mechanisms. However, conventional experimental methods to determine subcellular locations are usually time-consuming. The rapid development of high-throughput sequencing techniques has accelerated the demand for reliable and precise computational methods to locate apoptosis proteins’ subcellular positions from their primary sequences.

From the machine learning perspective, the identification of protein subcellular locations is usually described as a multi-class classification problem. Researchers have made great efforts in this field. These methods focus mainly on two aspects: (1) the construction of protein sequence encoding schemes and feature extraction; and (2) the design of a classification algorithm. There exist multiple machine learning techniques to estimate protein subcellular positions, such as covariant discriminant [2], fuzzy k-nearest neighbor [3,4], support vector machine (SVM) [5,6,7,8], and ensemble classifier [9,10]. Among these, SVM is widely used for its robust prediction performance. In addition, a series of feature extraction schemes has been developed to transform protein sequences into fixed-length numeric vectors, including amino acid composition (AAC) [11], pseudo-amino acid composition (PseAAC) [12,13,14,15,16], grouped weight encoding [17], wavelet coefficients [6], distance frequency [5], position-specific scoring matrix (PSSM) profile [18,19,20,21,22], and fusion of multi-view features [23,24]. 

Recently, SVM has demonstrated promising performance with a fast processing speed and has become the most popular classifier for researchers. The main difference among various SVM-based methods is the feature encoding schemes used. Among these, sequence representation models based on the PSSM profile have shown the most conspicuous improvements in the prediction accuracy aspect [25,26]. The PSSM profile of each sequence is usually achieved by executing the Position Specific Iterated BLAST (PSI-BLAST) program [27] against a specific protein database, e.g., NCBI’s non-redundant (NR) database or Swiss-Prot. Although the PSSM profile can provide important identifiable information for the prediction of protein subcellular location, several inherent limitations of alignment-based programs still exist. First, the PSI-BLAST technique is generally time-consuming and occupies more memory and has limited ability to deal with a large scale of sequence data. In addition, it is a challenge to construct a multiple sequence alignment, which is a type of NP-hard problem. Moreover, the accuracy of sequence alignments declines significantly due to the limited number of homologous sequences in the existing recognized database [28]. These issues have pushed us to design a more effective and alignment-free feature encoding method. 

In a previous study, a tri-gram encoding scheme was used to transform the PSSM profiles of proteins into 8000-dimensional feature vectors [18]. However, this method has two main shortcomings: (1) obtaining the PSSM profiles is usually time-consuming; and (2) high dimension data is more likely to cause the curse of dimensionality and costs too many computing resources. On the other hand, the physicochemical properties of amino acids are generally considered to affect the structure and function of proteins. In this study, we present an improved tri-gram encoding technique based on the protein overlapping property matrix (POPM) to lower the threshold of computing capability. The use of the scheme is as follows: First, 20 amino acids are divided into ten overlapping sets, where each group represents a distinctive physicochemical property. Second, each residue is encoded by using a 10-dimensional binary vector based on its physicochemical property. Third, the POPM of a protein consists of these row vectors related to amino acids at the corresponding positions in the sequence. Fourth, tri-gram encoding technique transforms the POPM into a 1000-dimensional feature vector. Finally, the optimal features after recursive feature selection are put into an SVM classifier to make predictions. The datasets used in this study and the source code for implementing the algorithm are freely available to the academic community at https://github.com/taigangliu/POPM-trigram.

## 2. Results and Discussion

### 2.1. Effects of Top K Features

After computing tri-gram features, a 1000-dimensional feature vector for each protein was obtained. Then, we acquired a ranked list of these features on the basis of their importance with the help of SVM-RFE. To find the ideal dimensions, the overall accuracies for the top *K* features were calculated by using jackknife cross-validation, where *K* = 10, 20, 30, ..., 300. We set the maximum value of *K* to be 300 because the prediction accuracies decline after reaching their peak points. 

Figure 1 shows the values on ZW225 and CL317 datasets with different top *K* features corresponding to their accuracies. It is clear that the overall accuracy (OA) for the ZW225 dataset reached the highest level when *K* climbed to 120. Besides, the CL317 dataset also obtained a favorable accuracy at this point. Therefore, we selected the top 120 features to represent a protein in the following study. Table 1 illustrates the performance of our method on two datasets by performing jackknife tests. As shown in the table, the accuracies of ZW225 and CL317 datasets reached relatively high levels of 98.2% and 96.2% respectively. Among these subcellular locations, the specificity (Spec) values were more than 98%, and the Matthews correlation coefficient (MCC) values were more than 92% for the two datasets. Notably, only the sensitivity (Sens) value of the secreted (Secr) location on the CL317 dataset was slightly lower than in the other locations and so was the accuracy of the mitochondrial (Mito) location on the ZW225 dataset. This may be due to the limited numbers of Mito and Secr proteins on the two datasets. Namely, the training sample size has an important influence on the accuracy.

### 2.2. Performance Comparison with Existing Methods

To assess the performance of our method objectively, we compared it with results from other existing methods based on the same datasets. The detailed outcomes of jackknife tests are reported in Table 2 and Table 3 where the Sens of each class and the OA are chosen as performance indexes. The comparison results of Spec and MCC of different methods on the two datasets are listed in Appendix A.

Based on Table 2, our method had an outstanding overall performance (98.2% in OA) on the ZW225 dataset, which was an improvement of over 10% compared with other methods such as Auto_Cova [12] and EN_FKNN [9]. Noticeably, the prediction accuracies of both Cyto proteins and Nucl proteins achieved 100%. Also, the Sens values of Memb and Mito reached relatively high prediction levels, with 96.6% and 96% respectively, which performed better than many other methods, including EBGW_SVM [17], DF_SVM [5], PSSM_AC [8], and ID_SVM [15]. In general, for the ZW225 dataset, our proposed method achieved a pleasing level.

In Table 3, for the CL317 dataset, our method generated a relatively high OA (96.2%) and achieved a remarkably enhanced performance for the subcellular locations of Cyto and Mito with 99.1% and 97.1% respectively. Compared with the previous study of the tri-gram PSSM algorithm [18], the proposed method not only improves the prediction accuracy but also largely reduces the computing costs. Admittedly, the PsePSSM-DCCA-LFDA [25] method performs excellently in every aspect, reaching 100% in almost all performance indexes for both datasets. This means that the combination of those three proven techniques—PsePSSM, DCCA, and LFDA—is effective for predicting protein subcellular locations. However, generating the PSSM profiles of query proteins by PSI-BLAST program is usually time-consuming and memory-consuming, which may limit its application with large-scale sequence data. To illustrate this point, the longest sequence (ID: Q68749, length: 3037) and the shortest sequence (ID: O43715, length: 76) of the datasets were selected to test the time required of this method. Remarkably, in our laboratory environment (Intel Xeon CPU E5620 @ 2.40GHz, 16 4-core processors, 16G RAM), it took 8334 seconds and 471 seconds to generate the PSSM profiles of two proteins (i.e., Q68749 and O43715) respectively. This result also indicates that the longer a sequence is, the more time it will take to process it. However, the required time for obtaining POPMs of two proteins is less than 1 second, which suggests that our method provides a convenient and fast way to extract features solely from amino acid sequences. The results also show that this relatively efficient method can achieve a favorable prediction accuracy as well.

In conclusion, our method not only greatly reduces the computational complexity but also obtains a comparable performance for predicting apoptosis protein subcellular locations. This significant progress is attributed to the powerful feature encoding scheme based on the tri-gram computed from POPM and SVM-RFE applied to select optimal features.

## 3. Materials and Methods

### 3.1. Datasets

Two benchmark datasets were employed to examine the validity of the proposed method: the ZW225 dataset [17] and the CL317 dataset [14,15]. The ZW225 dataset consists of 225 apoptosis proteins with 41 nuclear proteins, 70 cytoplasmic proteins, 25 mitochondrial proteins, and 89 membrane proteins. There are six types of subcellular locations in the CL317 dataset, including 112 cytoplasmic proteins, 47 endoplasmic reticulum proteins, 55 membrane proteins, 34 mitochondrial proteins, 52 nuclear proteins, and 17 secreted proteins. For short, cytoplasmic, membrane, mitochondrial, secreted, nuclear and endoplasmic reticulum are called Cyto, Memb, Mito, Secr, Nucl, and Endo, respectively.

### 3.2. Feature Extraction from POPM

It is demonstrated that, when properly represented, the amino acid composition of protein sequences contains the information necessary to delineate the structure and function of proteins. The physicochemical properties of amino acids encoded in protein sequences are believed to be important and original discriminatory information for predicting protein subcellular locations. 

In this work, Taylor’s overlapping physicochemical properties were adopted due to their successful application in catalytic residue prediction [29] and phosphorylation site identification [30]. The ten properties are listed in Table 4 [31]. Each amino acid residue was encoded using a 10-dimensional binary vector based on its physicochemical properties where the dimensions of the corresponding properties were set to 1 and the remaining positions were 0, e.g., A (0000100010) and V (0000110100). Thus, the POPM of a protein is defined as an *L* × 10 binary encoding matrix, which is denoted as [*M_i,j_*], where *i* = 1, 2, …, *L* denotes the position in the sequence and *j* = 1, 2, …, 10 denotes a physicochemical property. 

Then, the tri-gram encoding technique based on POPM was adopted to represent sequence samples, which reflected local interactions among three adjacent amino acids. Tri-gram features were generated using the following formula:(1)$$gram\left(x,y,z\right)=\frac{1}{L-2}\overset{L-2}{\underset{i=1}{\sum }}M_{i,x}\times M_{i+1,y}\times M_{i+2,z}\left(1\leq x,y,z\leq 10\right).$$

Hence, the total number of tri-gram features extracted from POPM was 1000. 

### 3.3. Support Vector Machine

SVM is a powerful machine learning model, which has been widely used for many protein prediction tasks in the field of computational biology [32,33,34,35,36,37]. Noticeably, this technique can construct an optimum hyperplane in a high-dimensional space to achieve precise linear classification. Besides, SVM can attain a non-linear classification with the use of a kernel trick. In this study, we chose the LIBSVM package [38] to help the SVM classifier work better. Among four in-built kernels provided by the LIBSVM package, i.e., linear, polynomial, radial basis function, and Gaussian, we adopted the linear kernel for this work, since it takes parameter optimization into account. 

### 3.4. Feature Selection by SVM-RFE

The contrasting dimensions of the feature vector can lead to obvious difference of efficiency in machine learning. The smaller the dimension, the less information it contains, which is not sufficient to identify the classification of the sample. In other words, the higher the dimension, the more information redundancy is involved, which not only greatly increases the computational complexity, but also affects the prediction accuracy. Therefore, we used support vector machine-recursive feature elimination (SVM-RFE) to select the appropriate features. This useful technique was originally applied for cancer classification [39] and then applied to predict functional attributes of proteins [40,41]. Firstly, we constructed a feature matrix of all the feature vectors of proteins according to each dataset, where each row represented a sample and each column corresponded to a feature. Next, all features were be ranked based on their importance through the SVM-RFE algorithm. Finally, the top *K* features ranked in the list were selected to represent a protein sequence.

### 3.5. Performance Evaluation

For statistical prediction, there are three types of cross-validation methods: the independent dataset test, the sub-sampling test, and the jackknife test [42,43]. In this research, the jackknife test was adopted to evaluate the performance of predictors because of its objectivity and rigorousness. During the jackknife test, each protein sequence in the dataset was picked out successively as a test sample, while the rest of protein sequences played the role of training samples.

To objectively assess the performance of our method, four standard performance indexes were reported, including Sens, Spec, and MCC for each subcellular location, and the OA [44,45]. They were defined using the following formulae:(2)$$Sens_{j}=\frac{TP_{j}}{TP_{j}+FN_{j}}=\frac{TP_{j}}{\left|C_{j}\right|},$$
(3)$$Spec_{j}=\frac{TN_{j}}{TN_{j}+FP_{j}}=\frac{TN_{j}}{\sum_{k\neq j}\left|C_{k}\right|},$$
(4)$$MCC_{j}=\frac{TP_{j}TN_{j}-FP_{j}FN_{j}}{\sqrt{\left(TP_{j}+FP_{j}\right)\left(TP_{j}+FN_{j}\right)\left(TN_{j}+FP_{j}\right)\left(TN_{j}+FN_{j}\right)}},$$
(5)$$OA=\frac{\sum_{j}TP_{j}}{\sum_{j}\left|C_{j}\right|}.$$
here, *TP_j_*, *TN_j_*, *FP_j_*, *FN_j_*, and |*C_j_*| indicate the number of true positives, true negatives, false positives, false negatives, and proteins in the subcellular location *C_j_*, respectively.

## 4. Conclusions

In this study, we focused on the design of a high-efficiency feature extraction technique for the prediction of the subcellular locations of apoptosis proteins. Firstly, a tri-gram encoding scheme based on POPM was introduced to transform the sequences of query proteins into 1000-dimensional feature vectors. Then, 120 optimal features selected by the SVM-RFE algorithm were input into a SVM prediction engine to perform the classification. The comparison with other existing models very strongly suggested that the proposed method is not encumbered by the limitations of alignment-based methods and could work as a very cost-effective tool for predicting subcellular location of apoptosis proteins. Due to the generality of this method, it is promising as an application for other protein classification problems in the future.

## Figures and Tables

**Figure 1 ijms-20-02344-f001:**
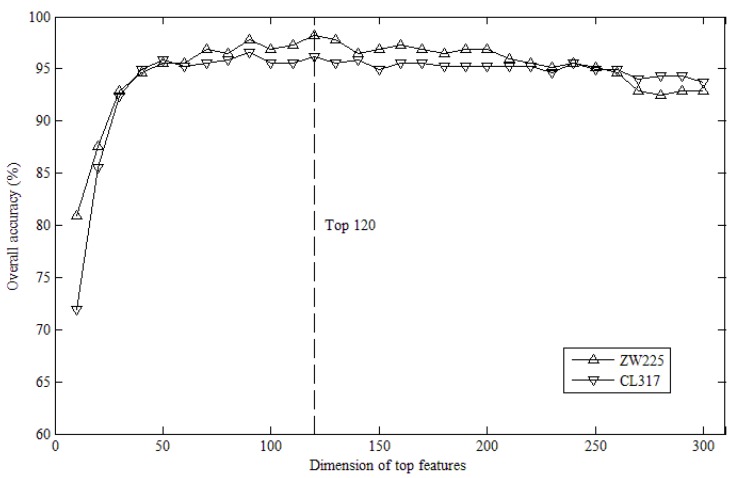
The graph illustrates the efficiency of various top features on the two datasets’ overall accuracies.

**Table 1 ijms-20-02344-t001:** Results for the two datasets by jackknife tests.

Dataset	Location ^1^	Sens (%)	Spec (%)	MCC	OA (%)
ZW225	Cyto	100	98.1	0.970	98.2
	Memb	96.6	100	0.972	
	Mito	96.0	100	0.977	
	Nucl	100	99.5	0.985	
CL317	Cyto	99.1	98.5	0.973	96.2
	Memb	92.7	98.9	0.923	
	Mito	97.1	99.3	0.951	
	Secr	88.2	100	0.936	
	Nucl	96.2	99.2	0.954	
	Endo	95.7	99.3	0.950	

^1^ For short, cytoplasmic, membrane, mitochondrial, secreted, nuclear and endoplasmic reticulum are called Cyto, Memb, Mito, Secr, Nucl, and Endo, respectively.

**Table 2 ijms-20-02344-t002:** Performance comparison of different methods on the ZW225 dataset.

Method	Sens for Each Class (%)				OA (%)
	Cyto	Memb	Mito	Nucl	
EBGW_SVM [17]	90.0	93.3	60.0	63.4	83.1
DF_SVM [5]	87.1	92.1	64.0	73.2	84.0
PSSM_AC [8]	82.9	92.1	68.0	78.0	84.0
ID_SVM [15]	92.9	91.0	68.0	73.2	85.8
Auto_Cova [12]	81.3	93.3	85.7	84.6	87.1
EN_FKNN [9]	94.3	94.4	60.0	80.5	88.0
Tri-gram PSSM [18]	97.1	98.9	96.0	97.6	97.8
PsePSSM-DCCA-LFDA [25]	100	98.9	100	100	99.6
Our method	100	96.6	96.0	100	98.2

**Table 3 ijms-20-02344-t003:** Performance comparison of different methods on the CL317 dataset.

Method	Sens for Each Class (%)						OA (%)
	Cyto	Memb	Mito	Secr	Nucl	Endo	
ID [14]	81.3	81.8	85.3	88.2	82.7	83.0	82.7
ID_SVM [15]	91.1	89.1	79.4	58.8	73.1	87.2	84.2
DF_SVM [5]	92.9	85.5	76.5	76.5	93.6	86.5	88.0
PseAAC_SVM [13]	93.8	90.9	85.3	76.5	90.4	95.7	91.1
PSSM-AC [8]	93.8	90.9	91.2	82.4	86.5	95.7	91.5
APSLAP [10]	99.1	89.1	85.3	88.2	84.3	95.8	92.4
Tri-gram PSSM [18]	98.2	96.4	94.1	82.4	96.2	95.7	95.9
PsePSSM-DCCA-LFDA [25]	99.1	100	100	100	100	100	99.7
Our method	99.1	92.7	97.1	88.2	96.2	95.7	96.2

**Table 4 ijms-20-02344-t004:** Amino acid groups based on Taylor’s overlapping properties.

Physicochemical Properties	Amino Acid Residues
Polar	N, Q, S, D, E, C, T, K, R, H, Y, W
Positive	K, H, R
Negative	D, E
Charged	K, H, R, D, E
Hydrophobic	A, G, C, T, I, V, L, K, H, F, Y, W, M
Aliphatic	I, V, L
Aromatic	F, Y, W, H
Small	P, N, D, T, C, A, G, S, V
Tiny	A, S, G, C
Proline	P

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
