# Peer review of "Predicting Apoptosis Protein Subcellular Locations based on the Protein Overlapping Property Matrix and Tri-Gram Encoding"

_ijms, 2019, doi:10.3390/ijms20092344_

Reviewer 1 Report

Yang and colleagues demonstrate a more accurate higher prediction of apoptosis protein subcellular location based on protein overlapping property matrix and tri-gram encodings. There are several issues with this paper that needs to be corrected before the paper can be considered for publication:

Generally, one can show higher accuracy while training machine learning (ML) classifier. The most important aspect of ML is the transferability of the prediction model. Here the authors have used two datasets (ZW225 and CL317) and considered both of them as a training dataset. I suggest them to use one as training and others as an independent dataset. While comparing method on the independent dataset, exclude the method that uses independent dataset as the training dataset. 

Furthermore, I suggest the authors construct the independent dataset and make sure it did not overlap with a training dataset of the existing methods. Finally, evaluate all the methods using independent dataset. I am sure showing 100% accuracy in Table2 and 3 is the result of overoptimization. 

Provide scripts and needed data so that the work can be reproduced by readers. Don't forget a license for the code/scripts.

What is the difference between tri-gram encoding utilized in this study and reference 18? This should be highlighted in the introduction section. 

Based on my experience, generally, RBF outperforms the other two kernels. Why did they select the linear kernel for this problem? I suggest the authors provide the parameter search range.

In page 7 “In the statistical prediction, there are three types of methods for cross-validation, which are independent dataset test, sub-sampling test, and jackknife test “. Reference is needed. I have listed a few here (PMID: 30649170 and 30425802)

In page 6 “SVM is a powerful machine learning model, which has been widely used for many protein prediction tasks in the field of computational biology.” I have listed few references (PMID: 30590410, 30239627, 29100375, 30081234, 29416743, 29868903). Please add more references.  

The authors should add references in many places. I have listed a few below:

In page 7, line 180. These metrics have been commonly used. It’s better to cite the recent papers.

No references for recursive feature elimination.

Author Response

Dear Reviewer 1:

We appreciate very much for your thoughtful comments and constructive suggestions, which help to improve the quality of our manuscript. Below, we provide a point-by-point response explaining how we have addressed each piece of your valuable advice. We have made the corrections accordingly in the revised manuscript, which are clearly highlighted. Thank you very much again for your attention to our paper.

Point 1: Generally, one can show higher accuracy while training machine learning (ML) classifier. The most important aspect of ML is the transferability of the prediction model. Here the authors have used two datasets (ZW225 and CL317) and considered both of them as a training dataset. I suggest them to use one as training and others as an independent dataset. While comparing method on the independent dataset, exclude the method that uses independent dataset as the training dataset.  

Response 1: Thank you for your instructive advice. In this study, the jackknife test is adopted to evaluate the prediction performance of our method. Although the jackknife test is time-consuming, it is considered more objective than other cross-validation methods (e.g. independent dataset test and sub-sampling test) [1]. The basic idea behind the jackknife test lies in systematically calculating the statistic estimate leaving out each sample from a dataset and then finding the average of these calculations. In addition, since there are different types of subcellular locations in the ZW225 and CL317 datasets, independent dataset test could not be applied directly. 

[1] Chou, K.C.; Zhang, C.T. Prediction of protein structural classes. Crit. Rev. Biochem. Mol. Biol. 1995, 30, 275-349

Point 2: Furthermore, I suggest the authors construct the independent dataset and make sure it did not overlap with a training dataset of the existing methods. Finally, evaluate all the methods using independent dataset. I am sure showing 100% accuracy in Table2 and 3 is the result of overoptimization.

Response 2: Thank you for your instructive advice. The independent dataset test is widely used to compare the performances of different predictive procedures. However, due to the limited number of experimentally validated apoptosis proteins in the database, the jackknife test is adopted instead by most of the existing methods. Also because of small samples, the prediction accuracies of several subcellular locations (e.g., Cyto proteins and Nucl proteins) achieve 100% in Tables 2 and 3. With increasing sample size, random variations of the effect are smaller and prediction results of these algorithms are more reliable. In addition, we shall make efforts in our future work to further estimate the generalization ability of our method by performing the independent dataset test.

Point 3: Provide scripts and needed data so that the work can be reproduced by readers. Don't forget a license for the code/scripts.

Response 3: Thank you for your instructive advice. The datasets used in this study and the source code for implementing the algorithm are freely available to the academic community at https://github.com/taigangliu/POPM-trigram.

Point 4: What is the difference between tri-gram encoding utilized in this study and reference 18? This should be highlighted in the introduction section.

Response 4: Thank you for your instructive advice. In this revision, we highlighted the differences between the present tri-gram encoding scheme and reference 18 in the introduction section as follows: “In the previous study, tri-gram encoding scheme was used to transform the PSSM profiles of proteins into 8000-dimensional feature vectors [18]. However, this method has two main shortcomings: (1) obtaining the PSSM profiles is usually time-consuming; (2) high dimension data is more likely to cause the curse of dimensionality and costs too many computing resources. On the other hand, the physicochemical properties of amino acids are generally considered to affect the structure and function of proteins. In this study, we present an improved tri-gram encoding technique based on protein overlapping property matrix (POPM) to lower the threshold of computing capability.”

Point 5: Based on my experience, generally, RBF outperforms the other two kernels. Why did they select the linear kernel for this problem? I suggest the authors provide the parameter search range.

Response 5: Thank you for your instructive advice. In this study, we prefer the linear kernel function to perform the SVM-RFE procedure since it can save the operation time. Indeed, preliminary test results indicate that RBF performs little better than the linear one, but meanwhile, RBF costs more time on parameter choice. In addition, to use the unified function in both feature selection and classification assessment, we choose the linear kernel for this work.

Point 6: In page 7 “In the statistical prediction, there are three types of methods for cross-validation, which are independent dataset test, sub-sampling test, and jackknife test”. Reference is needed. I have listed a few here (PMID: 30649170 and 30425802)

Response 6: Sorry for our negligence of the work. We read the two related papers carefully and cited them in this revision.

Point 7: In page 6 “SVM is a powerful machine learning model, which has been widely used for many protein prediction tasks in the field of computational biology.” I have listed few references (PMID: 30590410, 30239627, 29100375, 30081234, 29416743, 29868903). Please add more references. 

Response 7: Sorry for our negligence of the work. We read the above related papers carefully and cited them in this revision.

Point 8: The authors should add references in many places. I have listed a few below:

In page 7, line 180. These metrics have been commonly used. It’s better to cite the recent papers. No references for recursive feature elimination.

Response 8: Sorry for our negligence of the work. Thank you for your instructive advice. We read some recent papers carefully and cited them in this revision.

Reviewer 2 Report

The manuscript by Yang et al. presents a method to predict subcellular location of apoptosis proteins, based on amino acid chemico-physical properties and SVM classifier.
The new method proposed by the Authors performs well in comparison to most of the other methods presented in Tables 2 and 3. The tables compare sensitivity of the methods and overall accuracy. It should be interesting to compare also specificity and Matthews correlation coefficient, presented in table 1 for the method described in this manuscript.
Differences in the results of the two datasets should be commented in detail. Why the new method performs better with ZW225 than CL317 for Cyto/Memb/Nucl classes but worse for Mitocondrial proteins? Is the number of proteins in the datasets a limit for the correct assessment of the accuracy?
The Authors comment the better performance of the PsePSSM-DCCA-LFDA method when they discuss results on CL317 dataset, it should be correctly reported that PsePSSM-DCCA-LFDA performs better on both datasets. Moreover, the Authors claim that their method is fast, while the best performing method is time-consuming. This should be clearly shown by a table with a comparison of time required for the prediction, at least for these two methods, at best for all methods used in Tables 2 and 3. The time cost should be commented by considering the increasing computational power, from a perspective point of view: if a method is faster for a 10-fold factor, is it a detectable advantage for single user or just during the evaluation stage on a large data set ? If the reference time consists of hours, or minutes, I understand that improving the time is relevant. On the contrary, waiting for some more second to obtain prediction for a given protein, having a more accurate result, is not a real problem. The Authors quote in their manuscript the increasing time required for larger proteins, so the comparison of time needed should include also proteins of different sizes.
The Authors conclude (lines 121) that their method improves the prediction accuracy: this is wrong, because the best result is reached by an existing method.
The Authors write that the best results of the the PsePSSM-DCCA-LFDA method can be due to the combination of techniques. Is it possible to improve the results by combining these three techniques with the POPM technique?
English language and typos check needed. Examples:
line 36: "their" refers to proteins, so it must be "apoptosis proteins' function"
line 40: accelerates
line 48-49 a series ... has been
line 187: extraction
and so on.

Author Response

Dear Reviewer 2,

We highly appreciate and sincerely thank for your critical comments and professional advice! According to your instructive advice, we provided four additional tables with the other two indexes(i.e., specificity and MCC) as supplementary materials; supplemented an experiment to test the time required of these two methods for supporting our view; corrected the misleading and unclear expressions and so on. More specific responses are presented followed each valuable point you proposed. Thank you very much again for your attention to our paper.

Point 1: The new method proposed by the Authors performs well in comparison to most of the other methods presented in Tables 2 and 3. The tables compare sensitivity of the methods and overall accuracy. It should be interesting to compare also specificity and Matthews correlation coefficient, presented in table 1 for the method described in this manuscript. 

Response 1: Thank you for your instructive advice. Although the usual performance indexes most articles choose are sensitivity and overall accuracy (OA), we fully agree that adding other two indexes you recommended, specificity (Spec) and Matthews correlation coefficient (MCC), can compare our method with others more comprehensively. The comparison results of Spec and MCC of different methods on the two datasets are listed in Tables S1-S4 in supplementary materials.

Point 2: Differences in the results of the two datasets should be commented in detail. Why the new method performs better with ZW225 than CL317 for Cyto/Memb/Nucl classes but worse for Mitocondrial proteins? Is the number of proteins in the datasets a limit for the correct assessment of the accuracy?

Response 2:Thank you for your instructive advice. It is true that the number of training sample size have an important influence on accuracy. In this revision, we added comments accordingly as follows: “Notably, only the Sens value of Secr location on the CL317 dataset is slightly lower than the other locations. And so is the accuracy of Mito location on the ZW225 dataset. This may be due to the limited numbers of Mito and Secr proteins on the two datasets. Namely, the number of training sample size has an important influence on accuracy.”

Point 3: The Authors comment the better performance of the PsePSSM-DCCA-LFDA method when they discuss results on CL317 dataset, it should be correctly reported that PsePSSM-DCCA-LFDA performs better on both datasets.

Response 3: Thank you for your instructive advice. Sorry for our loose expression and we have made corrections accordingly in this revision as follows: “Admittedly, PsePSSM-DCCA-LFDA [25] method performs excellently in every aspect, reaching 100% in almost all performance indexes for both datasets.”

Point 4: Moreover, the Authors claim that their method is fast, while the best performing method is time-consuming. This should be clearly shown by a table with a comparison of time required for the prediction, at least for these two methods, at best for all methods used in Tables 2 and 3. The time cost should be commented by considering the increasing computational power, from a perspective point of view: if a method is faster for a 10-fold factor, is it a detectable advantage for single user or just during the evaluation stage on a large data set ? If the reference time consists of hours, or minutes, I understand that improving the time is relevant. On the contrary, waiting for some more second to obtain prediction for a given protein, having a more accurate result, is not a real problem. The Authors quote in their manuscript the increasing time required for larger proteins, so the comparison of time needed should include also proteins of different sizes.

Response 4: Thank you for your instructive advice. In this revision, we supplemented an experiment to support our view. The longest sequence (ID: Q68749, Length: 3037) and the shortest sequence (ID: O43715, Length: 76) of the datasets are selected to test the time required of these two methods.

Remarkably, in our laboratory environment (Intel Xeon CPU E5620 @ 2.40GHz, 16 4-core processors, 16G RAM), it takes 8334 seconds and 471 seconds to generating the PSSM profiles of two proteins (i.e., Q68749 and O43715) respectively. This result also indicates that the longer one sequence is, the more time it will take. However, the required time for obtaining POPMs of two proteins is less than 1 second, which suggests that our method provides a convenient and fast way to extract features solely from amino acid sequences.

Point 5: The Authors conclude (lines 121) that their method improves the prediction accuracy: this is wrong, because the best result is reached by an existing method.

Response 5: Thank you for your instructive advice. Sorry for our negligence of the work and misnomer. In this revision, we have made corrections accordingly as follows: “In conclusion, our method not only greatly reduces the computational complexity but also obtains comparable performance for predicting apoptosis protein subcellular location.”

Point 6: The Authors write that the best results of the PsePSSM-DCCA-LFDA method can be due to the combination of techniques. Is it possible to improve the results by combining these three techniques with the POPM technique?

Response 6: Thank you for your instructive advice. The combination of POPM technique and PsePSSM-DCCA-LFDA method shall improve results to a predictable extent. However, obtaining the PSSM profiles is usually time-consuming and memory-consuming and thus is of limited use with large-scale sequence data. In this study, we focus mainly on the design of a relatively time-saving and high-quality protein encoding scheme. POPM technique meets this expectation and results  also suggest that the physicochemical properties of amino acids encoded in protein sequences could serve as an important and original discriminatory information for predicting protein subcellular location. In addition, we shall make efforts in our future work to further improve prediction performance by combining POPM and other powerful techniques.

Point 7: English language and typos check needed. Examples:

line 36: "their" refers to proteins, so it must be "apoptosis proteins' function"

line 40: accelerates

line 48-49 a series ... has been

line 187: extraction

and so on.

Response 7:  Thank you for your instructive advice. Sorry for our negligence of the work. We have proofread our manuscript carefully, and some grammar mistakes have been corrected in the revised manuscript.

Round  2

Reviewer 1 Report

The authors have satisfactorily responded to all my questions and made the necessary changes to the manuscript. Therefore, I strongly recommend this paper for the publication.

Reviewer 2 Report

No more comments